# The Predictive Value of the Fibrinogen–Albumin-Ratio Index on Surgical Outcomes in Patients with Advanced High-Grade Serous Ovarian Cancer

**DOI:** 10.3390/cancers16193295

**Published:** 2024-09-27

**Authors:** Magdalena Postl, Melina Danisch, Fridolin Schrott, Paul Kofler, Patrik Petrov, Stefanie Aust, Nicole Concin, Stephan Polterauer, Thomas Bartl

**Affiliations:** Department of Obstetrics and Gynecology, Division of General Gynecology and Gynecologic Oncology, Medical University Vienna, 1090 Vienna, Austria; magdalena.postl@meduniwien.ac.at (M.P.); melina.danisch@meduniwien.ac.at (M.D.); n01442859@students.meduniwien.ac.at (F.S.); paul.kofler@campus.lmu.de (P.K.); pp.petrov@gmx.at (P.P.); stefanie.aust@meduniwien.ac.at (S.A.); nicole.concin@meduniwien.ac.at (N.C.); thomas.bartl@meduniwien.ac.at (T.B.)

**Keywords:** ovarian cancer, HGSOC, tumor load, cytoreductive surgery, neoadjuvant chemotherapy, FARI, fibrinogen, albumin

## Abstract

**Simple Summary:**

The optimal selection of patients with advanced high-grade serous ovarian cancer (HGSOC) who are likely to benefit from upfront cytoreductive surgery remains challenging. With the results of the TRUST trial, the fifth randomized phase III trial comparing oncologic outcomes of primary versus interval cytoreductive surgery, expected to be presented soon, there is a high unmet clinical need to define novel biomarkers to optimize pretherapeutic patient assessment and allow for more personalized clinical decision making. Building on the growing evidence of the predictive and prognostic value of fibrinogen in various solid cancers, this study is the first to observe an independent association between the “Fibrinogen–Albumin-Ratio Index” (FARI) and surgical outcomes in patients with advanced HGSOC undergoing primary cytoreductive surgery. Given that the FARI seems to act as a surrogate for intra-abdominal tumor load, the further clinical validation of this cost-effective and readily available biomarker appears promising.

**Abstract:**

Background/Objectives: The present study evaluates predictive implications of the pretherapeutic Fibrinogen–Albumin-Ratio Index (FARI) in high-grade serous ovarian cancer (HGSOC) patients undergoing primary cytoreductive surgery. Methods: This retrospective study included 161 patients with HGSOC International Federation of Gynecology and Obstetrics (FIGO) stage ≥ IIb, who underwent primary cytoreductive surgery followed by platinum-based chemotherapy. Associations between the FARI and complete tumor resection status were described by receiver operating characteristics, and binary logistic regression models were fitted. Results: Higher preoperative FARI values correlated with higher ascites volumes (r = 0.371, *p* < 0.001), and higher CA125 levels (r = 0.271, *p* = 0.001). A high FARI cut at its median (≥11.06) was associated with lower rates of complete tumor resection (OR 3.13, 95% CI [1.63–6.05], *p* = 0.001), and retrained its predictive value in a multivariable model independent of ascites volumes, CA125 levels, FIGO stage, and Charlson Comorbidity Index (CCI). Conclusions: The FARI appears to act as a surrogate for higher intra-abdominal tumor load. After clinical validation, FARI could serve as a readily available serologic biomarker to complement preoperative patient assessment, helping to identify patients who are likely to achieve complete tumor resection during primary cytoreductive surgery.

## 1. Introduction

Cytoreductive surgery with the primary goal of complete macroscopic tumor resection is the mainstay of treatment in advanced ovarian cancer. Complete macroscopic tumor resection has been highlighted to be one of the most important prognostic factors [1]. Lately, four randomized phase III trials comparing primary cytoreductive surgery followed by chemotherapy to neoadjuvant chemotherapy followed by interval cytoreductive surgery did not observe significant survival differences between either approach; a fifth trial is currently underway [2,3,4,5,6]. Considering less radical interventions and less surgical morbidity following interval cytoreductive surgery, selected patients may benefit from neoadjuvant chemotherapy depending on tumor load and spread of the disease [2,3,4,5]. The scientific debate on the optimal selection of patients for either primary or interval cytoreductive surgery, however, is still ongoing. 

Evidence on predictive biomarkers that may help to complement preoperative patient assessment and predict the outcome of primary cytoreductive surgery remains limited. The latest update of the 2024 ESGO-ESMO-ESP consensus conference highlights that no validated biomarkers predictive of the success of surgical resection are available to date, addressing the unmet clinical need for further research [7,8].

An increasing body of evidence indicates that systemic coagulation and inflammation are closely associated with tumor development and progression [9]. Among various coagulation factors, fibrinogen appears to play a pivotal role as elevated plasma fibrinogen levels are associated with a poor prognosis in different cancer types, including ovarian cancer [10,11,12,13,14]. Fibrinogen, a glycoprotein synthesized in hepatocytes, is converted by thrombin to fibrin, binds platelets, supports platelet aggregation, and ultimately promotes blood clotting. Besides its critical role in the coagulation cascade, fibrinogen belongs to a group of “positive” acute-phase proteins and is expressed in response to systemic inflammation [15]. Conversely, albumin belongs to a group of “negative” acute-phase proteins, and decreased serum levels typically indicate inflammation processes [16,17]. Furthermore, albumin serves as a parameter to assess the nutritional status of a patient. Low albumin serum levels are indicative of malnutrition, which is associated with various clinical implications, such as lower quality of live, higher risk of therapy-associated side effects, reduced response to chemotherapy, and reduced cancer survival [18].

Recently, the Fibrinogen–Albumin-Ratio Index (FARI), which considers both fibrinogen and albumin levels, was proposed as a novel prognostic biomarker in several cancer types [17,19,20,21,22,23,24]. In ovarian cancer, a high FARI was previously associated with a reduced progression-free (PFS) and overall survival (OS) in clear cell ovarian cancer; Yu et al. reported the FARI to predict resistance to neoadjuvant chemotherapy in advanced epithelial ovarian cancer patients irrespective of specific histologic subtypes. However, potential associations between the FARI and surgical outcomes, such as complete tumor resection in high-grade serous ovarian cancer (HGSOC) patients, have not been evaluated to date [25,26,27].

The present study therefore aims to assess both predictive implications of the pretherapeutic FARI on surgical outcomes and its prognostic value in HGSOC patients undergoing primary cytoreductive surgery.

## 2. Materials and Methods

The present study was designed as a single-center retrospective chart review evaluating data of all consecutive patients with advanced stage HGSOC International Federation of Gynecology and Obstetrics (FIGO) stage ≥ IIb, who underwent primary cytoreductive surgery followed by platinum-based chemotherapy at the Medical University of Vienna between 2006 and 2020.

Exclusion criteria from the final analysis included neoadjuvant chemotherapy, a diagnosis of another cancer type within five years, incomplete preoperative laboratory results including missing fibrinogen and albumin levels, and less than 12 months of oncologic follow-up at our department (Figure 1).

Data on pretherapeutic peripheral venous serum fibrinogen and albumin levels were retrieved from archived routine blood test results, which were performed 21 days prior to primary surgery as part of our department’s clinical routine. As previously reported, all preoperative laboratory tests were performed as part of an ongoing routine preoperative assessment including albumin, fibrinogen, and CA125 from a peripheral venous puncture. Serum albumin levels and fibrinogen levels were quantified using citrated plasma. Albumin assays were performed with bromocresol green using routine clinical chemical photometric analyzers [28,29]. Fibrinogen levels were quantified using clotting reagents and the Clauss method [30,31]. CA125 levels were quantified by an electrochemiluminescence immunoassay using blood serum [32]. Thereafter, the FARI was calculated as the ratio of fibrinogen (g/L) divided by albumin (g/L) and multiplied by 100 as previously described [21,23].

Cytoreductive surgery was performed only by European Society of Gynaecological Oncology (ESGO)-accredited gynecologic oncologists and included median laparotomy and maximal cytoreductive surgery with routine upper abdominal exploration. Standard adjuvant cytotoxic therapy consisted of six cycles of a platinum-based chemotherapy doublet +/− bevacizumab and/or poly ADP ribose polymerase (PARP) inhibitor maintenance in case of either BRCA mutation or homologous recombination deficiency (HRD), as indicated by our department’s internal tumorboard.

After the completion of cytotoxic chemotherapy +/− maintenance therapy, if applicable, patients were invited to the institutions’ internal follow-up program, which includes routine clinical examinations, serologic tests (CA125), and imaging studies. Follow-up examinations are recommended four times annually for three years, followed by two examinations until year 5 and annual examinations until a maximum of 10 years.

A statistical analysis was performed using the Statistical Package for the Social Sciences statistical software (IBM SPSS Statistics Version 29.0.0.0 for MAC) and R: a language and environment for statistical computing (R Core Team 2024, R-4.3.3. Angel Food Cake). Patient baseline characteristics were analyzed by descriptive statistics. Categorical variables were described as absolute frequencies and percentages; continuous variables are given as medians and interquartile ranges. For further assessment, we decided to cut the FARI at its median instead of an optimized cut-off to avoid model overfitting. In line with this, we also chose to apply the median of CA125 to allow for better comparison. To identify potential associations between the FARI and clinicopathological variables, the independent-samples T-test, independent-samples Mann–Whitney U test, chi-squared test, one-way ANOVA, and spearman’s correlation coefficient were used, where appropriate. To evaluate a potential association of the FARI and surgical outcomes, receiver operating characteristics (ROCs) were calculated for descriptive purposes and binary regression models were fitted applying cut variables to simulate clinical applicability. In parallel to the FARI, univariate and multivariable models were also built for fibrinogen and albumin alone to allow for the descriptive contextualization of respective predictive values. Survival analyses were performed for the endpoints of PFS and disease-specific survival (DSS) to validate the observations of the impact of the FARI on complete tumor resections. Of note, multivariable models were fitted for descriptive purposes only to assess whether associations of the FARI with either endpoint may be interpreted as independent, or, on the contrary, may act as a surrogate of another variable. The follow-up data cut-off was on 31 June 2023. PFS was defined as the time between the date of the diagnosis and the date of the last follow-up or the date of the first recurrence as documented by the department’s internal tumorboard. DSS was defined as time from the date of the diagnosis to the date of the last follow-up or date of cancer-related death. For survival analyses, both univariate and multivariable Cox regression models were fitted. Patient survival was depicted by Kaplan–Meier curves. To account for the limited sample size of the present cohort and to avoid overfitting, clinically meaningful established parameters were chosen as potential confounders as previously suggested for prognostic modeling [33]. Survival was censored on the last date they were known to be alive. For all statistical tests, *p*-values below 0.05 were considered statistically significant. To account for multiple testing, a Bonferroni correction was applied to the binary regression models testing the primary hypothesis, increasing the statistical significance threshold to *p* < 0.0042 (Table 1). Odds and hazard ratios were indicated with a two-sided confidence interval of 95%, where appropriate. Prior to initiation, this study was approved by the Institutional Review Board of the Medical University of Vienna (IRB approval number: 1383/2018). 

## 3. Results

### 3.1. Descriptive Characteristics

In total, 161 patients were included in the final statistical analysis. A high FARI ≥ 11.06 cut at its median was associated with a lower rate of complete tumor resection (*p* = 0.001). Calculated as a continuous variable, the FARI correlated positively with preoperative CA125 levels (r = 0.271, *p* = 0.001) and higher ascites volumes (r = 0.371, *p* < 0.001). Detailed pretherapeutic patients’ characteristics are provided in Table 2.

### 3.2. The FARI Is Independently Associated with Complete Tumor Resection during Primary Cytoreductive Surgery 

A high FARI ≥ 11.06 (OR 3.13, 95% CI [1.63–6.05], *p* = 0.001) was associated with a lower rate of complete tumor resections and retained its predictive value (OR 2.90, 95% CI [1.41–5.97], *p* = 0.004) in a multivariable model as compared to the FIGO stage, presence of ascites, preoperative CA125 levels, and CCI (Table 1). Both fibrinogen ≥ 4.46 g/L (OR 2.93, 95% CI [1.53–5.63], *p* = 0.001) and albumin ≤ 39.9 g/L (OR 3.00, 95% CI [1.56–5.76], *p* = 0.001) were also associated with a lower rate of complete tumor resection after primary cytoreductive surgery (Table 1). Fibrinogen and albumin, however, did not retain their respective predictive values in a multivariable model as compared to the FIGO stage, presence of ascites, preoperative CA125 levels, and CCI after applying a Bonferroni correction (Appendix A). An ROC analysis was calculated for descriptive purposes only, yielding an AUC of the FARI for the endpoint of complete tumor resection of 0.688 (0.605–0.771), as compared to 0.654 (0.568–0.740) for fibrinogen alone and 0.656 (0.570–0.742) for albumin alone.

### 3.3. The FARI Is Independently Associated with PFS and DSS

The FARI, calculated as a continuous variable, was associated with both impaired PFS (HR 1.07, 95% CI [1.04–1.10], *p* < 0.001) and DSS (HR 1.07, 95% CI [1.03–1.10], *p* < 0.001) in univariate Cox regression. Kaplan–Meier curves including confidence intervals depicting significantly decreased PFS and DSS in patients with a preoperatively high FARI cut at its median are shown in Figure 2a,b. The FARI retained its predictive value in multivariable analyses as compared to the FIGO stage, complete tumor resection, and CCI (PFS: HR 1.06, 95% CI [1.02–1.09], *p* = 0.002; DSS: HR 1.07, 95% CI [1.03–1.11], *p* = 0.001). Further details are provided in Appendix A. Three-year PFS rates for low- and high-FARI groups were 33.8% and 17.3%.

### 3.4. Subgroup Analyses of Patients with Complete Tumor Resection

A subgroup analysis of patients, for whom complete tumor resection could be achieved during primary cytoreductive surgery, was performed to confirm the independent effect of the FARI as compared to complete tumor resection status. Within the subgroup of patients with complete tumor resection after primary cytoreductive surgery (*n* = 95, 59.0%), the FARI calculated as a continuous variable retained its predictive value for both PFS (HR 1.07, 95% CI [1.03–1.12], *p* = 0.001) and DSS (HR 1.09, 95% CI [1.04–1.14], *p* < 0.001). Both associations could be confirmed in a multivariable model as compared to CCI and FIGO (PFS: HR 1.08, 95% CI [1.03–1.12], *p* = 0.001; DSS: HR 1.10, 95% CI [1.05–1.15], *p* < 0.001).

## 4. Discussion

A high pretherapeutic FARI is associated with a lower rate of complete tumor resection during primary cytoreductive surgery. This observation appears to be independent of the FIGO stage, preoperative CA125 levels, presence of ascites, and CCI. In line with this, the FARI is also associated with both impaired PFS and DSS; this effect remains stable in multivariable analyses as compared to the FIGO stage, tumor resection status, and CCI. Of note, the FARI retains its prognostic value in a subgroup analysis of patients with complete tumor resection during primary cytoreductive surgery. 

Three studies have previously investigated the prognostic implication of fibrinogen and albumin in ovarian cancer. Chen et al. evaluated the FARI in 114 patients with clear cell ovarian cancer and showed that a high FARI was an independent risk factor for platinum resistance and was associated with impaired survival rates. The FARI was also associated with complete tumor resection; however, it was not investigated if the FARI has any predictive effect [25]. Yu et al. calculated the preoperative albumin-to-fibrinogen ratio in 313 patients with advanced epithelial ovarian cancer, irrespective of histologic subtypes, who were treated with neoadjuvant chemotherapy followed by interval cytoreductive surgery. In line with our findings, there was a significant association between a low albumin-to-fibrinogen ratio and reduced PFS and OS. A low albumin-to-fibrinogen ratio was also associated with a lower rate of complete tumor resection; however, the albumin-to-fibrinogen ratio was not evaluated as a predictive biomarker for complete tumor resection [26]. Li et al. investigated the outcome of 186 patients with epithelial ovarian cancer, irrespective of histologic subtypes, and divided them into three groups based on their fibrinogen–albumin score. The fibrinogen–albumin score was able to predict DSS; however, Li et al. did not find any statistically significant association between the fibrinogen–albumin score and tumor size [27].

A potential association between high fibrinogen and low albumin levels was previously observed for several other gynecological malignancies. In stage IB-IIA cervical cancer patients, who received radical hysterectomy followed by adjuvant chemotherapy as primary treatment, an increased FARI was associated with impaired survival [34]. Huang et al. reported a correlation between a low albumin-to-fibrinogen ratio and lymph node metastasis, distant metastasis, depth of stromal infiltration, tumor size, and FIGO stage [35]. In breast cancer patients, the FARI seems to have a prognostic implication, especially in the stage II/III subgroup and in the luminal A-like subtype [36]. A Chinese report combined the FARI with a platelet–lymphocyte ratio score and reported a correlation with survival rates in 707 breast cancer patients [37]. A recently published report on the FARI in vulvar cancer interestingly showed an association between a low FARI and worse survival rates in patients undergoing primary surgical resection [23].

The impact of elevated fibrinogen and low albumin levels on survival in cancer patients has been evaluated in several studies based on their roles in reflecting the systemic inflammatory response and nutritional status, both of which are closely associated with cancer progression and patient outcomes [9,10,11,12,13,14,15]. Hypoalbuminemia appears to be indicative of malnutrition and chronic inflammation, with are both conditions that are prevalent in advanced cancer and correlate with a poor prognosis [16,17,18]. Therefore, albumin has been incorporated in several prognostic scores, for example, the Royal Marsden Hospital (RMH) score, including albumin levels, lactate dehydrogenase (LDH) levels, and the number of metastases. A high RMH score has been shown to be associated with impaired progression-free and overall survival in several cancer types; however, as far as we are aware, the RMH score has not been evaluated in ovarian cancer patients yet [38]. Elevated fibrinogen levels, on the other hand, are associated with cancer-related hypercoagulability and inflammation, promoting tumor growth, metastasis, and reduced survival rates [15]. Fibrinogen has been further reported as a prognostic factor not only in oncologic diseases but also in several other serious medical conditions [39,40,41].

In ovarian cancer patients, Polterauer et al. showed that higher fibrinogen levels were associated with impaired overall and progression-free survival [14]. Those findings were confirmed by Luo et al., who reported that fibrinogen even outperformed the prognostic value of CA125 levels [42]. Seebacher et al. presented fibrinogen as a potential predictor of ovarian malignancy in patients with adnexal masses [43]. Looking at albumin as a prognostic factor in ovarian cancer, Ataseven et al. has shown that pretherapeutic hypoalbuminemia was associated with a higher rate of postoperative complications and impaired overall survival [44]. Low albumin levels seem to be associated with a poor prognosis in patients either undergoing upfront cytoreductive surgery or primarily receiving neoadjuvant chemotherapy [45,46].

Previous randomized phase III trials could not demonstrate significant survival differences when comparing oncologic outcomes of advanced HGSOC patients following primary versus interval cytoreductive surgery. Despite this, the optimal patient selection for either approach remains controversial, and no validated algorithm that reliably identifies patients who would benefit rather well from either primary or interval cytoreductive surgery is available to date [2,3,4,5]. The decision if complete tumor resection can be achieved during primary cytoreductive surgery is currently entirely based on surgical and / or imaging assessment [47]. Identifying potential biomarkers, which support clinical decision making of which treatment strategy should be pursued, is of high clinical interest. Regarding the potential predictive value of CA125 and complete tumor resection, our study was not set out to assess any potential associations; however, to our knowledge, the previous literature on a potential predictive value of CA125 and complete tumor resection during primary cytoreductive surgery remains controversial, and one study for secondary cytoreductive surgery is negative [48,49]. The FARI thereby appears to be a promising biomarker, as the FARI seems to predict complete tumor resection independent of the FIGO tumor stage, preoperative CA125 levels, presence of ascites, and CCI before surgery. As we observed significant associations between a high FARI and the presence of ascites, as well as a high FARI and higher levels of preoperative CA125, it could be hypothesized that the FARI potentially acts as a surrogate for more extensive intra-abdominal tumor load. This hypothesis is supported by previous observations that CA125 levels and fibrinogen levels positively correlate with tumor burden, whereas lower albumin levels indicate cachexia along with malnutrition [14,32,45,50]. To date, preoperatively quantifying intra-abdominal tumor load remains challenging, and the most common categorization of stage FIGO IIIc may encompass a broad range of intra-abdominal involvement. Therefore, the FARI could serve as a promising biomarker to complement preoperative patient assessment, which could ultimately help to improve patient selection for either primary or interval cytoreductive surgery.

The present study faces several limitations. First, since this is a retrospective study, a potential patient selection bias or incomplete data acquisition might distort the clinical value of the FARI. Second, due to the long timeframe of this study, patients received different therapy regimes due to change in treatment standards, especially regarding maintenance therapy; and data on HRD status were only available for patients treated after 2017. Third, we had no prospective information on any subjective peritoneal cancer index, such as the Fagotti Score, to quantify intra-abdominal tumor spread, which would have been interesting to further support associations between the FARI and tumor load [51,52]. Fourth, the FARI was assessed only before surgery; therefore, no statement can be made as to whether or not its postoperative course may improve its prognostic value. To account for potential limitations regarding maintenance therapy and HRD status, data on maintenance regimens were provided. Given the low number of administered PARP inhibitors, and balanced bevacizumab administrations, any related meaningful bias appears unlikely. To assess whether the FARI solely predicts survival as a surrogate for complete tumor resection, subgroup analyses were conducted, making this scenario unlikely. The FARI obviously shows very close correlations with both fibrinogen and albumin levels; and the present study was not set out to assess the additional diagnostic value as compared to either parameter alone. However, a slightly higher ROC-AUC in favor of FARI, and the fact that both fibrinogen and albumin failed to retain statistical significance in multivariable testing following a Bonferroni correction, indicates a slightly higher predictive value of the FARI as compared to either value alone; however, in the case of the future validation of the FARI, a comparative assessment of additional diagnostic values appears reasonable.

## 5. Conclusions

As the preoperative FARI appears to act as a surrogate for intra-abdominal tumor load, clinical validation as a promising predictor of complete tumor resection in patients with advanced HGSOC undergoing primary cytoreductive surgery appears promising. The FARI could complement the pretherapeutic patient assessment and selection of patients who are likely to benefit from primary cytoreductive surgery. 

## Figures and Tables

**Figure 1 cancers-16-03295-f001:**
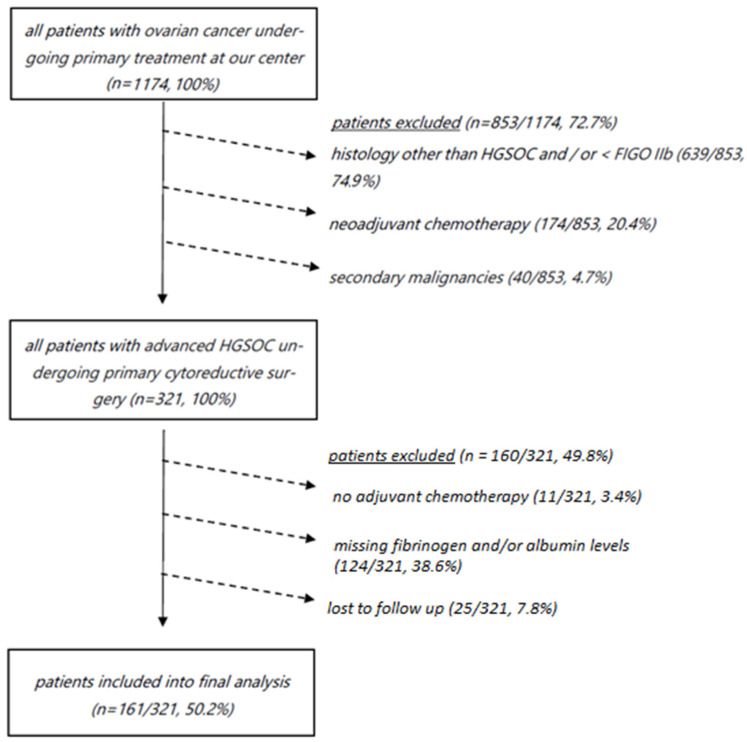
Consort diagram depicting all patients with ovarian cancer undergoing primary treatment who were not included into final analysis (*n* = 1013/1174, 86.3%). HGSOC, high-grade serous ovarian cancer; FIGO, International Federation of Gynaecology and Obstetrics.

**Figure 2 cancers-16-03295-f002:**
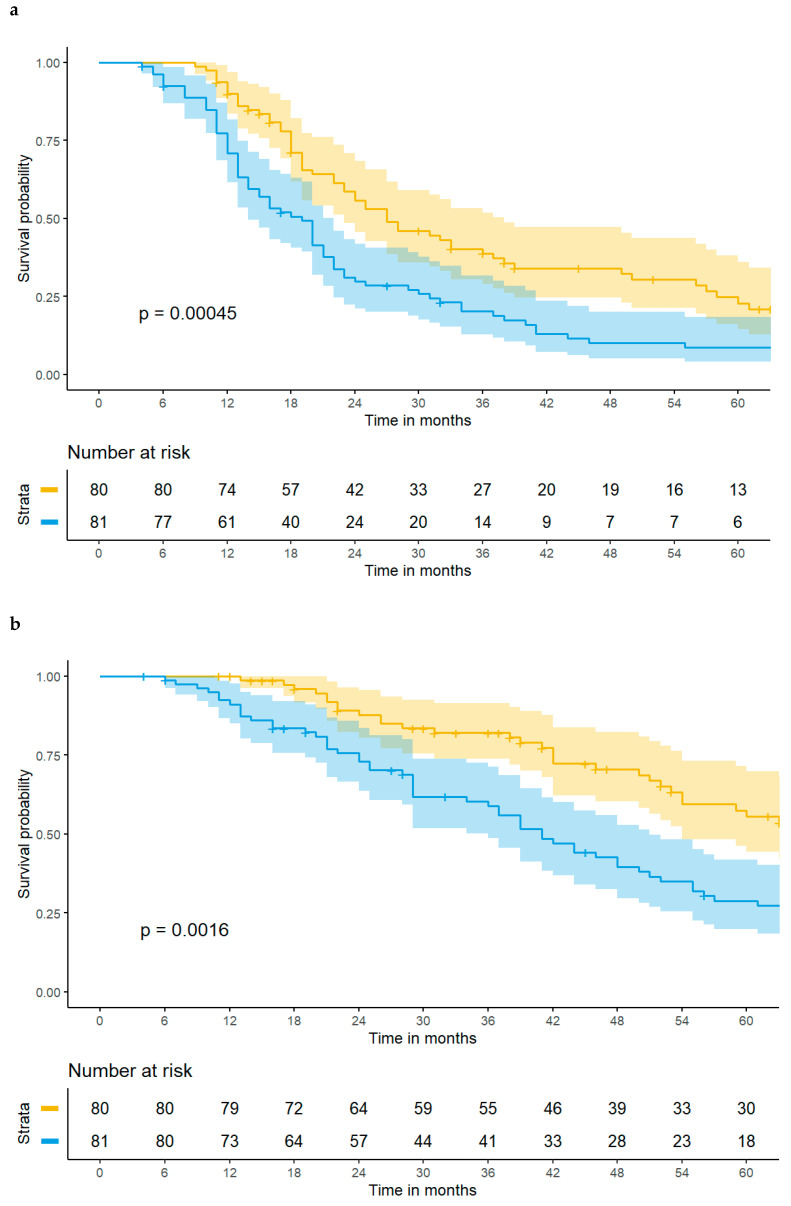
(**a**) A high FARI ≥ 11.06 is associated with impaired progression-free survival in patients with advanced high-grade serous ovarian cancer. The yellow line depicts patients with a FARI < 11.06 and the blue line depicts patients with a FARI ≥ 11.06, with 95% confidence intervals, respectively. (**b**) A high FARI ≥ 11.06 is associated with impaired disease-specific survival in patients with advanced high-grade serous ovarian cancer. The yellow line depicts patients with a FARI < 11.06 and the blue line depicts patients with a FARI ≥ 11.06, with 95% confidence intervals, respectively.

**Table 1 cancers-16-03295-t001:** Predictive value of the FARI in a univariable and multivariable binary logistic regression model with the endpoint of complete tumor resection during primary cytoreductive surgery in advanced high-grade serous ovarian cancer patients.

	Univariable		Multivariable	
Parameter	*p*-Value	OR (95% CI)	*p*-Value	OR (95% CI)
FARI ≥ 11.06	0.001 ^a^	3.13 (1.63–6.05)	0.004 ^b^	2.90 (1.41–5.97)
CCI ≥ 3	0.906 ^a^	0.95 (0.43–2.11)	0.792 ^b^	0.89 (0.60–5.52)
CA125 ≥ 683.0 kU/L	0.383 ^a^	1.33 (0.70–2.50)	0.621 ^b^	0.83 (0.40–1.72)
FIGO stage IIb-IIIc vs. IV	0.592 ^a^	1.32 (0.48–3.62)	0.286 ^b^	1.83 (0.60–5.52)
ascites ≤ 50 mL/51–499 mL/≥500 mL	0.010 ^a^	1.60 (1.12–2.30)	0.109 ^b^	1.38 (0.93–2.04)
fibrinogen ≥ 4.46 g/L	0.001 ^a^	2.93 (1.53– 5.63)	-	-
albumin ≤ 39.9 g/L	0.001 ^a^	3.00 (1.56–5.76)	-	-

Preoperative fibrinogen and albumin are depicted for descriptive purposes only to contextualize the predictive value of the FARI. After applying a Bonferroni correction, a *p* < 0.0042 was considered as statistically significant. HGSOC, high-grade serous ovarian cancer; CCI, Charlson Comorbidity Index; FIGO, International Federation of Gynaecology and Obstetrics; OR, odds ratio; CI, confidence interval. ^a^ Univariable binary logistic regression model; ^b^ multivariable binary logistic regression model.

**Table 2 cancers-16-03295-t002:** Descriptive characteristics of patients with advanced high-grade serous ovarian cancer undergoing primary cytoreductive surgery broken down by the FARI cut at its median.

Parameter	*n* (%) or Median (IQR)			*p*-Value
	Overall Cohort	FARI < 11.06	FARI ≥ 11.06	
number of patients	161 (100%)	80 (49.7%)	81 (50.3%)	
age at therapy (years)	58.0 (50.0–67.0)	58.0 (50.5–68.0)	58.0 (50.0–65.5)	0.812 ^a^
ECOG				1.000 ^b^
0	148 (91.9%)	74 (92.5%)	74 (91.4%)	
1	13 (8.1%)	6 (7.5%)	7 (8.6%)	
CCI	4.0 (3.0–5.0)	4.0 (3.0–5.0)	4.0 (3.0–5.0)	0.260 ^a^
preoperative CA125 levels (kU/l)	683.0 (177.8–1513.0)	295.0 (105.2–965.2)	910.5 (354.7–2034.5)	<0.001 ^c^
FIGO stage (2009)				0.078 ^b^
IIb–IIIc	144 (89.4%)	68 (85.0%)	76 (93.8%)	
IV	17 (10.6%)	12 (15.0%)	5 (6.2%)	
complete tumor resection				0.001 ^b^
yes	95 (59.0%)	58 (72.5%)	44 (54.3%)	
no	66 (41.0%)	22 (27.5%)	37 (45.7%)	
ascites				<0.001 ^d^
≤50 mL	58 (36.0%)	38 (47.5%)	20 (24.7%)	
51–499 mL	25 (15.5%)	16 (20.0%)	9 (11.1%)	
≥500 mL	78 (48.4%)	26 (32.5%)	52 (64.2%)	
reoperation within the 30-day postoperative period				0.803 ^b^
yes	17 (10.6%)	9 (11.3%)	8 (9.9%)	
no	144 (89.4%)	71 (88.8%)	73 (90.1%)	
time between surgery and first cycle of adjuvant chemotherapy (days)	24.0 (20.0–28.0)	24.5 (21.0–28.0)	23.0 (18.3–27.8)	0.081 ^a^
recurrence				0.003 ^b^
yes	128 (79.5%)	56 (70.0%)	72 (88.9%)	
no	33 (20.5%)	24 (30.0%)	9 (11.1%)	
adjuvant bevacizumab				0.637 ^b^
yes	81 (50.3%)	42 (52.5%)	42 (51.9%)	
no	80 (49.7%)	38 (47.5%)	39 (48.1%)	
adjuvant PARP inhibitor				0.985 ^d^
yes	6 (3.7%)	3 (3.8%)	3 (3.7%)	
no	151 (93.8%)	75 (93.8%)	76 (93.8%)	
unknown *	4 (2.5%)	2 (2.5%)	2 (2.5%)	
status at last follow-up				0.002 ^d^
no evidence of disease	39 (24.2%)	28 (35.0%)	11 (13.6%)	
stable disease	13 (8.1%)	7 (8.8%)	6 (7.4%)	
progression	22 (13.7%)	10 (12.5%)	12 (14.8%)	
intercurrent death	7 (4.3%)	3 (3.8%)	4 (4.9%)	
cancer-related death	80 (49.7%)	32 (40.0%)	48 (59.3%)	
preoperative fibrinogen levels (g/L)	4.4 (3.7–5.5)	3.7 (3.1–4.0)	5.5 (4.8–6.3)	<0.001 ^a^
preoperative albumin levels (g/L)	40.1 (36.3–43.5)	43.2 (41.1–45.7)	36.7 (33.8–39.1)	<0.001 ^a^

HGSOC, high-grade serous ovarian cancer; CCI, Charlson Comorbidity Index; ECOG, Eastern Cooperative Oncology Group Performance Status; FIGO, International Federation of Gynaecology and Obstetrics; PARP inhibitor, poly (ADP-Ribose) polymerase inhibitor. ^a^ Independent-samples T-test; ^b^ chi-squared test; ^c^ independent-samples Mann–Whitney U test; ^d^ one-way ANOVA. * Patients included in the DUO-O study, blinded.

## Data Availability

Data of this retrospective study are available on request from the corresponding author.

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
