# Peer review of "The Predictive Value of the Fibrinogen–Albumin-Ratio Index on Surgical Outcomes in Patients with Advanced High-Grade Serous Ovarian Cancer"

_cancers, 2024, doi:10.3390/cancers16193295_

Round 1

Reviewer 1 Report

Comments and Suggestions for Authors

The article discusses the predictive value of the Fibrinogen-to-Albumin Ratio Index (FARI) in surgical outcomes in patients with advanced high-grade serous ovarian cancer. I would like to offer the following points for the authors to consider for improvement of the article:

1-The methodology needs to be more explicitly detailed, particularly on how FARI is calculated and its comparative performance against other biomarkers. How cut-off values ​​for FARI are determined and validated should be included.

2- The article should include a discussion of other recent prognostic scores that include albumin. This can be expanded in the “Discussion” section to compare and contrast FARI with other established scores such as the Royal Marsden Hospital (RMH) Score (https://doi.org/10.3390/cancers16101835).

3- The discussion should expand on the biological rationale for using albumin and fibrinogen as biomarkers. Discuss how these markers interact with tumor biology, particularly in the context of inflammation and cancer cachexia, and support the use of FARI as a prognostic tool. This will provide a stronger theoretical basis for the study's findings.

4- Although the study applied Bonferroni correction, it is critical to ensure that statistical analyses are robust, particularly in multivariate models. Additional analyses, such as sensitivity analyses or the use of bootstrapping methods, may increase the reliability of the findings.

Comments on the Quality of English Language

Moderate editing of English language required

Author Response

The article discusses the predictive value of the Fibrinogen-to-Albumin Ratio Index (FARI) in surgical outcomes in patients with advanced high-grade serous ovarian cancer. I would like to offer the following points for the authors to consider for improvement of the article:

We highly appreciate your time and effort to assess our manuscript and tried to incorporate your feedback in our manuscript.

Comment 1: The methodology needs to be more explicitly detailed, particularly on how FARI is calculated and its comparative performance against other biomarkers. How cut-off values ​​for FARI are determined and validated should be included.

Response 1: As previously described the FARI is calculated as the ratio of fibrinogen (g/L) divided by albumin (g/L) and multiplied by 100 (line 118-120). Since there is no validated standard value of the FARI, the FARI was cut at its median. To assess a potential additional predictive value of the FARI, we compared to both fibrinogen and albumin, indicating a slightly higher predictive value of the FARI as compared to either value alone.

Comment 2: The article should include a discussion of other recent prognostic scores that include albumin. This can be expanded in the “Discussion” section to compare and contrast FARI with other established scores such as the Royal Marsden Hospital (RMH) Score (https://doi.org/10.3390/cancers16101835).

Response 2: We agree that it is worth mentioning that albumin has been included in other prognostic scores. Therefore, we added the following sentence in the discussion (line 294-299):

“Therefore, albumin has been incorporated in several prognostic scores, for example the Royal Marsden Hospital (RMH) score, including albumin levels, lactate dehydrogenase (LDH) levels, and number of metastases. A high RMH score has been shown to be as-sociated with impaired progression free and overall survival in several cancer types; however as far as we are aware of the RMH score has not been evaluated in ovarian cancer patients yet.”

Comment 3: The discussion should expand on the biological rationale for using albumin and fibrinogen as biomarkers. Discuss how these markers interact with tumor biology, particularly in the context of inflammation and cancer cachexia, and support the use of FARI as a prognostic tool. This will provide a stronger theoretical basis for the study's findings.

Response 3: We agree that we should expand on the biological rationale for using albumin and fibrinogen as biomarkers. Therefore, we have added the following paragraph in the discussion section (line 281-303).

The impact of elevated fibrinogen and low albumin levels on survival in cancer patients has been evaluated in several studies based on their roles in reflecting the sys-temic inflammatory response and nutritional status, both of which are closely associated with cancer progression and patient outcomes. Hypoalbuminemia appears to be indicative of malnutrition and chronic inflammation, with are both conditions that are prevalent in advanced cancer and correlate with poor prognosis. Therefore, al-bumin has been incorporated in several prognostic scores, for example the Royal Marsden Hospital (RMH) score, including albumin levels, lactate dehydrogenase (LDH) levels, and number of metastases. A high RMH score has been shown to be associated with impaired progression free and overall survival in several cancer types; however as far as we are aware of the RMH score has not been evaluated in ovarian cancer patients yet. Ele-vated fibrinogen levels, on the other hand, are associated with cancer-related hyperco-agulability and inflammation, promoting tumor growth, metastasis, and reduced survival rates. Fibrinogen has been further reported as a prognostic factor not only in oncologic diseases but also in several other serious medical conditions.”

Comment 4: Although the study applied Bonferroni correction, it is critical to ensure that statistical analyses are robust, particularly in multivariate models. Additional analyses, such as sensitivity analyses or the use of bootstrapping methods, may increase the reliability of the findings.

Response 4: We may thank the reviewer for raising this very valid concern. We want to highlight that building multivariable models to assess associations with complete tumor resection (binary logistic regression) and PFS/DSS (cox regression) was by no means meant to present a functional model to apply in clinical routine (which would have required at least an independent validation cohort), but to demonstrate that potential associations of FARI with either endpoint appear to be independent of clinically relevant variables. To clarify this, we added following sentence to the methods section:

“Of note, multivariable models were fitted for descriptive purposes only to assess whether associations of FARI with either endpoint may be interpreted as independent, or, in the contrary, may act as a surrogate of another variable.” (line 152-155)

We agree that assuring the quality of presented models is of utmost importance, therefore we:

  1. Applied a Hosmer-Lemeshow Test to the multivariable binary regression mode to confirm its goodness-of fit (p=0.398).
  2. Performed bootstrapping on both the univariable binary regression of FARI and the multivariable regression of FARI. Following 1000 bootstrap replicates, results remained stable, univariable bootstrapped FARI on R0: p=0.003, 95%CI 1.60-5.90; multivariable bootstrapped FARI on PFS: p=0.003, 95%CI 1.42-6.86
  3. Performed bootstrapping on both the univariable Cox-regression of FARI and the multivariable Cox-regression of FARI on PFS. Following 1000 bootstrap replicates, results remained stable, univariable bootstrapped FARI on PFS: p=0.001, 95%CI 1.01-3.04; multivariable bootstrapped FARI on PFS: p=0.009, 95%CI 1.01-1.12.
  4. Performed bootstrapping on both the univariable Cox-regression of FARI and the multivariable Cox-regression of FARI on DSS. Following 1000 bootstrap replicates, results remained stable, univariable bootstrapped FARI on DSS: p=0.010, 95%CI 1.02-1.14; multivariable bootstrapped FARI on DSS: p=0.014, 95%CI 1.00-1.14

As, in summary, bootstrapping confirmed the seemingly independent associations of FARI with either endpoint, and, did not relevantly alter the CIs, we would consider that presented models support the observations of the present manuscript. We may include bootstrapping results in results supplement on the reviewer’s behest.

Reviewer 2 Report

Comments and Suggestions for Authors

This paper investigates the prognostic  role of FARI in patients udergoing primary citoreductive surgery for high serous ovarian cancer.

1)It is relevant to report in the introduction  that fibrinogen has been reported as  a prognostic factor not only in oncology but in several other diseases as acute cardiovascular diseases. A study was done on this, and some cautions should be considered when discussing the prognostic role. A study has been performed on this issue :Ferraro S, Santagostino M, Marano G, Colli E, Vendramin C, Maffé S, Rossi L, Galvani M, Panteghini M, Bongo AS. The prognostic value of plasma fibrinogen concentrations of patients with ST-elevation myocardial infarction and treated by primary percutaneous coronary intervention: A cautionary message. Scand J Clin Lab Investig 2012;72:355–62

2) The laboratory methods to measure CA125, Fibrinogen and Albumin have not been reported in the methods. If for each biomarker several methods have been used, indicate the coefficient of variability between methods(available in the laboratory).

3)in the table the unit of measurement of Ca125 has not been reported

4)the cut-off of Ca125>600 is not clear as it it has been determined. Why the Roc curve for Ca125 has not been determined? Ca125 is the main prognostic marker for Ovarian Cancer monitoring and this should be reported in the introduction. it is very strange your result, have you any explanation?You can find useful this study reporting about the relevance of CA125 monitoring to evaluate the relapse:Ferraro S, Robbiano C, Tosca N, Panzeri A, Paganoni AM, Panteghini M. Serum human epididymis protein 4 vs. carbohydrate antigen 125 in ovarian cancer follow-up. Clin Biochem 2018;60:84-90.

5) There are some mistakes"an ROC"

Comments on the Quality of English Language

moderate revision

Author Response

We highly appreciate your time and effort to assess our manuscript and tried to incorporate your feedback in our manuscript.

Comment 1: It is relevant to report in the introduction that fibrinogen has been reported as a prognostic factor not only in oncology but in several other diseases as acute cardiovascular diseases. A study was done on this, and some cautions should be considered when discussing the prognostic role. A study has been performed on this issue :Ferraro S, Santagostino M, Marano G, Colli E, Vendramin C, Maffé S, Rossi L, Galvani M, Panteghini M, Bongo AS. The prognostic value of plasma fibrinogen concentrations of patients with ST-elevation myocardial infarction and treated by primary percutaneous coronary intervention: A cautionary message. Scand J Clin Lab Investig 2012;72:355–62

Response 1: Thank you for your valuable feedback. We do agree that it is important to mention that Fibrinogen acts as a prognostic factor not only in oncology. After extensive review of literature, we decided to add the following sentence “Fibrinogen has been further reported as a prognostic factor not only in oncologic diseases but also in several other serious medical conditions” in the discussion section (line 301-303) and quote three metanalyses.

Comment 2: The laboratory methods to measure CA125, Fibrinogen and Albumin have not been reported in the methods. If for each biomarker several methods have been used, indicate the coefficient of variability between methods (available in the laboratory).

Response 2: To clarify the laboratory methods we have added the following paragraph (line 111-118).

“As previously reported, all preoperative lab tests were performed as part of an ongoing routine preoperative assessment including albumin, fibrinogen and CA125 from a peripheral venous puncture. Serum albumin levels and fibrinogen levels were quantified using citrated plasma. Albumin assays were performed with bromocresol green using routine clinical chemical photometric analyzers [a] [b]. Fibrinogen levels were quantified using clotting reagents and the Clauss method [c] [d]. CA124-levels were quantified by electrochemiluminescence immunoassay using blood serum. [e]”

a: Hill PG. The measurement of albumin in serum and plasma. Ann Clin Biochem. 1985;22(Pt 6):565–578. doi: 10.1177/000456328502200604.

b: Bekos C, Polterauer S, Seebacher V, Bartl T, Joura E, Reinthaller A, Sturdza A, Horvat R, Schwameis R, Grimm C. Pre-operative hypoalbuminemia is associated with complication rate and overall survival in patients with vulvar cancer undergoing surgery. Arch Gynecol Obstet. 2019 Oct;300(4):1015-1022. doi: 10.1007/s00404-019-05278-7.

c: CLAUSS A. Gerinnungsphysiologische Schnellmethode zur Bestimmung des Fibrinogens [Rapid physiological coagulation method in determination of fibrinogen]. Acta Haematol. 1957 Apr;17(4):237-46. German. doi: 10.1159/000205234.

d: Bekos C, Grimm C, Brodowicz T, Petru E, Hefler L, Reimer D, Koch H, Reinthaller A, Polterauer S, Polterauer M. Prognostic role of plasma fibrinogen in patients with uterine leiomyosarcoma - a multicenter study. Sci Rep. 2017 Nov 3;7(1):14474. doi: 10.1038/s41598-017-13934-8.

e: Hasholzner U, Baumgartner L, Stieber P, Meier W, Hofmann K, Fateh-Moghadam A. Significance of the tumour markers CA 125 II, CA 72-4, CASA and CYFRA 21-1 in ovarian carcinoma. Anticancer Res. 1994 Nov-Dec;14(6B):2743-6.

Comment 3: in the table the unit of measurement of Ca125 has not been reported

Response 3: Thank you for pointing this out; we added the measurement of CA125 to the table.

Comment 4: the cut-off of Ca125>600 is not clear as it it has been determined. Why the Roc curve for Ca125 has not been determined? Ca125 is the main prognostic marker for Ovarian Cancer monitoring and this should be reported in the introduction. it is very strange your result, have you any explanation?You can find useful this study reporting about the relevance of CA125 monitoring to evaluate the relapse:Ferraro S, Robbiano C, Tosca N, Panzeri A, Paganoni AM, Panteghini M. Serum human epididymis protein 4 vs. carbohydrate antigen 125 in ovarian cancer follow-up. Clin Biochem 2018;60:84-90.

Response 4: Thank you for your valuable comment. There is no previous data on the prognostic and predictive value of the FARI und this specific research questions. For the FARI we decided to use the median instead of an optimize cut off to avoid model over fitting. In line we also chose to apply the median of CA125 to allow for better comparison. The cut-off for CA125 was chosen based on the median which was 683 kU/L in the overall cohort. To underline this we mentioned this in the method section (line 140-142). 

Regarding the potential predictive value of CA125 and complete tumor resection, our study was not set out to assess any potential associations; however, to our knowledge previous literature on a potential predictive value of CA125 and complete tumor resection during primary cytoreductive remains controversial, one study for secondary cytoreductive surgery is negative. To clarify this point we added a sentence in the discussion (line 322-327).

Comment 5: There are some mistakes"an ROC"

Response 5: Thank you for pointing out this spelling mistake, we made the correction.

Reviewer 3 Report

Comments and Suggestions for Authors

The authors present the results of their retrospective study of the association between “Fibrinogen-Albumin-Ratio Index” (FARI) and surgical outcomes in patients with advanced HGSOC undergoing primary cytoreductive surgery."

They conclude "Given the FARI seems to act as a surrogate for intraabdominal tumor load, further clinical validation 21 of this cost-effective and readily available biomarker appears promising."

Overall, the topic is timely and of substantial interest, the study was well-designed, and the manuscript well-written.  The authors were careful to remain within the appropriate boundaries of a retrospective study.

My primary comment/suggestion concerns the Discussion Section.  While the authors present a brief overview of fibrinogen and albumin in their introduction and an excellent comparison to the published literature in their discussion, the manuscript would be strengthened by additional discussion of the potential reasons WHY fibrinogen and albumin levels are important in cancer patients.  In other words, in addition to the excellent literature review/comparison, please explain WHY.

Comments on the Quality of English Language

Very minor editing for typos, language and grammar needed

Author Response

We highly appreciate your time and effort to assess our manuscript and tried to incorporate your feedback in our manuscript.

Response 1: We agree that we should expand on the biological rationale for using albumin and fibrinogen as biomarkers. Therefore, we have added the following paragraph in the discussion section (line 281-303).

The impact of elevated fibrinogen and low albumin levels on survival in cancer patients has been evaluated in several studies based on their roles in reflecting the sys-temic inflammatory response and nutritional status, both of which are closely associated with cancer progression and patient outcomes. Hypoalbuminemia appears to be indicative of malnutrition and chronic inflammation, with are both conditions that are prevalent in advanced cancer and correlate with poor prognosis. Therefore, al-bumin has been incorporated in several prognostic scores, for example the Royal Marsden Hospital (RMH) score, including albumin levels, lactate dehydrogenase (LDH) levels, and number of metastases. A high RMH score has been shown to be associated with impaired progression free and overall survival in several cancer types; however as far as we are aware of the RMH score has not been evaluated in ovarian cancer patients yet. Ele-vated fibrinogen levels, on the other hand, are associated with cancer-related hyperco-agulability and inflammation, promoting tumor growth, metastasis, and reduced survival rates. Fibrinogen has been further reported as a prognostic factor not only in oncologic diseases but also in several other serious medical conditions.”

Round 2

Reviewer 1 Report

Comments and Suggestions for Authors

I am satisfied that the authors have addressed all of my previous concerns about the article. It is now much improved and I feel that it is now suitable for publication.

Comments on the Quality of English Language

I am satisfied that the authors have addressed all of my previous concerns about the article. It is now much improved and I feel that it is now suitable for publication.

Author Response

Comment 1: I am satisfied that the authors have addressed all of my previous concerns about the article. It is now much improved and I feel that it is now suitable for publication.

Response 1: Thank you for the time and effort dedicated to providing valuable feedback on our manuscript.

Reviewer 2 Report

Comments and Suggestions for Authors

Comment 1: in the introduction a background on Ca125 as prognostic marker in OvCa monitoring should be better stated. You can use the reference below which is a recent paper on therapeutic monitoring

Comment 2 : The immunoassay for CA125 detection has not been reported. Please clarify.

You quote a reference of 1997, but different generations of CA125 immunoassays have passed since 1997! The paper I have suggested to you reports data on OVCa monitoring with a current immunoassay [Ferraro S, Robbiano C, Tosca N, Panzeri A, Paganoni AM, Panteghini M. Serum human epididymis protein 4 vs. carbohydrate antigen 125 in ovarian cancer follow-up. Clin Biochem 2018;60:84-90]. I suggest to consider this reference instead of previous one.

Comment 3:  When you define a cut-off/threshold the rationale for selecting this threshold should be well explained or the evaluation of the predictive power of the marker may be wrong. A threshold  value you report seems too high. The median of the distribution is not a good reason to choose this value as a cut-off.

Comments on the Quality of English Language

extensive editing required, there are still some mistakes

Author Response

We may thank the reviewer for the time and effort, which already greatly contributed to the quality and comprehensibility of the paper. Please find point-by-point answers as follows:

Comment 1: in the introduction a background on Ca125 as prognostic marker in OvCa monitoring should be better stated. You can use the reference below which is a recent paper on therapeutic monitoring

We thank the reviewer for the valuable comment. We agree that further information on potential markers relevant for the present study should be included to complement the introduction. As, however, CA125 is undoubtedly one of the most important prognostic markers, its predictive value remains controversial and a matter of discussion. Following these considerations and the reviewer’s previous remarks, we are convinced that the reviewer will agree to discuss the value of the CA125 in the discussion section, not in the introduction (discussion: lines 631-644). To account of the reviewer’s present remark, we added a paragraph on predictive markers in the intro. As CA125 is specifically mentioned in the 2024 ESGO-ESMO-ESP consensus conference recommendations, we decided to not further expand this topic, as it is not the primary focus of the present manuscript.

Evidence on predictive biomarkers which may help to complement preoperative patient assessment and predict the outcome of primary cytoreductive surgery remains limited. The latest update of the 2024 ESGO-ESMO-ESP consensus conference highlights that no validated biomarkers predictive of the success of surgical resection are available to date, addressing the unmet clinical need for further research. (lines 53-57)

Comment 2 : The immunoassay for CA125 detection has not been reported. Please clarify.
You quote a reference of 1997, but different generations of CA125 immunoassays have passed since 1997! The paper I have suggested to you reports data on OVCa monitoring with a current immunoassay [Ferraro S, Robbiano C, Tosca N, Panzeri A, Paganoni AM, Panteghini M. Serum human epididymis protein 4 vs. carbohydrate antigen 125 in ovarian cancer follow-up. Clin Biochem 2018;60:84-90]. I suggest to consider this reference instead of previous one.

We thank the reviewer of the remark - Elecsys CA 125 II by Roche was used for CA125 detection at our institution until the year 2020/21. The information was added in the methods section / line 112).

We previously quoted Hasholzner et al. as one of the first reference papers on the role of CA125 in ovarian cancer. However, we agree, that citing the newest and most comprehensive literature may be more useful. We thank the reviewer for the literature suggestion, but after careful literature review, we decided to quote Charkhchi et al, Cancers 2020, who do not only extensively cover the topics of different immunoassays and monitoring (in turn also citing papers of the workgroup of Ferraro et al), but also providing a comprehensive literature overview.

Comment 3:  When you define a cut-off/threshold the rationale for selecting this threshold should be well explained or the evaluation of the predictive power of the marker may be wrong. A threshold  value you report seems too high. The median of the distribution is not a good reason to choose this value as a cut-off.

We share the reviewer’s concerns on the selection of a threshold. As abundantly discussed in literature, defining an “universal” cut-off is a Sisyphean task, which is impossible to resolve, a respective cut-off would always be dependent on the baseline data sample, i.e. respective patient cohorts at different centers.

However, the present study was not set out to propose a specific clinical cut-off but was rather meant to exploratorily point out that FARI may contain valuable predictive information regarding surgical outcomes, which could help clinical decision making in future. As there is no established data on promising FARI thresholds available to date, a cut-off may be created either by 1) choosing an arbitrary threshold following the data distribution, e.g. the median, or 2) to apply an outcome-based “optimized” cut-off, with the obvious risk of overfitting. We chose the first approach, as it is more conservative, and the probably most widely applied method for exploratory predictive/prognostic research. We would assume these considerations as fundamental basics  for all readers of predictive/prognostic research papers, and therefore decided not to repeat this over again, as this would go way beyond the scope of the present paper.

To address the reviewer’s concerns and support the validity of our observations, we:

  • repeated calculations of the binary logistic regression with the endpoint if macroscopic complete resection in a multivariable model using the same variables as given in table 2 but using continuous variables (which would not reflect clinical reality but increase statistical power to dispel potential doubts on artificial model overfitting). Results remain stable, with FARI being independently predictive of macroscopic complete resection (FARI: OR 1.10 95%CI [1.02-1.19], p=0.013).
  • repeated calculations of the binary logistic regression with the endpoint if macroscopic complete resection in a multivariable model after defining an optimized cut of using a Youden’s J based on the ROC-AUC of FARI vs. R0/1. Interestingly, the optimized FARI cut-off is very close to the median (10.10 vs. 11.06). Applying the optimized FARI, results of the regression model (fitted using dichotomized covariates) remained stable, with FARI being independently predictive of macroscopic complete resection (FARI: OR 4.18 95%CI [1.93-9.07], p=0.013).

Even though we share the reviewer’s concerns, we think that dwelling on cut-off values and respective methods of identification will greatly increase the length of the paper without increasing its scientific value, especially as defining a specific cut-off for future research was not the intended goal of the manuscript. We may offer to discuss the choice of an arbitrary vs. optimized cut offs in predictive/prognostic research by the example of present FARI data in a method supplement on the editor’s behest.

Reviewer 3 Report

Comments and Suggestions for Authors

The authors adequately addressed the issue I raised in my initial review.  Excellent paper!

Author Response

Comment 1: The authors adequately addressed the issue I raised in my initial review.  Excellent paper!

Response 1: Thank you for the time and effort dedicated to providing valuable feedback on our manuscript.